# Study on Carbon Fixation Ratio and Properties of Foamed Concrete

**DOI:** 10.3390/ma16093441

**Published:** 2023-04-28

**Authors:** Yuansheng Wei, Xiaoqiang Cao, Gang Wang, Mingguang Zhang, Zhiwen Lv

**Affiliations:** 1College of Safety and Environmental Engineering, Shandong University of Science and Technology, Qingdao 266590, China; caoxiaoqiang@sdust.edu.cn (X.C.); gang.wang@sdust.edu.cn (G.W.); 13598091558@163.com (Z.L.); 2Mine Disaster Prevention and Control-Ministry of State Key Laboratory Breeding Base, Shandong University of Science and Technology, Qingdao 266590, China

**Keywords:** solid waste-based materials, foam concrete, carbon dioxide, sequestration, pore structure

## Abstract

Using solid waste to sequester carbon dioxide not only reduces the greenhouse effect but also reuses resources. However, the existing solidified carbon dioxide storage materials are expensive and have poor storage effect. Therefore, in this study, cement, solid waste base material, and 30% hydrogen peroxide were used to make foamed concrete materials through chemical foaming, and XRD, BET, SEM, and thermogravimetric techniques were used to explore the amount of carbon dioxide adsorbed by foamed concrete materials under different ratio conditions. The results show that (1) the hydration products of the cementified materials mainly include C-S-H, Ht and Ca(OH)_2_, which are important factors for the storage of CO_2_. (2) A water–cement ratio of 0.7 and a foaming agent dosage of 10% are the best ratios for foamed concrete materials. With the increase of the water–cement ratio and the dosage of the foaming agent, the amount of CO_2_-sealed stock first increases and then decreases. (3) The maximum carbon dioxide sealing capacity of foamed concrete material is 66.35 kg/m^3^.

## 1. Introduction

Climate change is a great challenge related to human survival and long-term development. Low-carbon development has increasingly become one of the keys to sustainable human development. Therefore, how to reduce the CO_2_ in the atmosphere and overcome the global greenhouse effect has attracted great attention globally.

CO_2_ capture is an effective method to reduce the CO_2_ in the atmosphere. Many capture methods, such as absorption [1], adsorption [2], membrane separation [3], hydrate-based separation [4], and cryogenic distillation [5], have been developed. The captured CO_2_ can further be reutilized in many fields, such as in the production of ammonia and urea, food and beverages, refrigerants, and fire extinguishing gases [6,7]. However, the reutilized CO_2_ only accounts for 2% of the emissions [8], which is a tiny contribution to CO_2_ reduction. In addition, most methods still exhibit shortcomings in terms of application convenience and cost. The cement-based materials can capture CO_2_ from the atmosphere during the hydration reaction [9,10], and thus their application potentials in CO_2_ sequestration have gradually attracted great attention [11,12,13]. Pure cement materials, such as cement and concrete, can sequester CO_2_ but not cost-efficiently. Therefore, the composition and preparation process of cement material needs to be optimized for the economic cost of the implementation and carbon footprint issue [14]. The CO_2_ diffusion rate is the controlling step for the early carbonation reaction rate of cement material [15,16]. Carbonation reduces the porosity of the carbonized area and forms carbonates at the edges of the cement particles, which inhibits CO_2_ diffusion. Due to the compact structure of concrete, the carbonation depth is small. Therefore, only a very small amount of the components reacts with CO_2_.

Based on the analysis above, cementitious materials prepared from solid wastes (slags, carbide slags, and mining solid wastes) can be further foamed and cured into porous materials to realize large-scale CO_2_ capture [17]. Relevant studies have shown that various solid wastes, such as bauxite tailings [18], red mud [19], gold tailings [20], fly ash [21], silt [22], and so on, can be used as the raw material for this purpose. Cai investigated the effects of lime and fly ash contents on the compressive strength of foam concrete and found both of the foam concretes obtained with 10% lime mix ratio and 20% fly ash mix ratio exhibited the highest 28-day compressive strength [23]. The strength and pore structure of the porous material can also be tuned by adjusting the components, admixtures, and reaction process during the preparation. To study the properties of fly ash, Łach et al. prepared foam concrete with fly ash substituted for cement, using water, glass, and sodium hydroxide as the activators and 30% hydrogen peroxide as the foaming agent [24].

The foam concrete exhibited a dry density of 400–600 kg/m^3^, the highest compressive strength of 3.4 MPa, and the lowest thermal conductivity of 0.0826 W/(m·k). Chen [25] was able to regulate the pore structure and strength of porose material by adjusting the ratio of raw materials and the type of admixtures during the preparation process. Xi studied the effects of the raw material ratio on pore structure, and the results showed that the porosity decreased with the increase of water–cement ratio and fly ash content and increased with the increase of water–cement ratio [26]. The influence of the water–cement ratio was most significant, and the equivalent average pore size was positively correlated with these three parameters. Zhang et al. reported the optimal raw material ratio for the preparation of foam concrete by a physical foaming method using titanium-containing waste residue and red gypsum as the admixtures, which was ordinary Portland (PO) cement:titanium-containing waste residue:red gypsum = 10:45:45 (mass ratio), 2% lime, and 4% sulfoaluminate cement [27]. The dry density and 28-day compressive strength of the obtained foam concrete reached up to 437 kg/m^3^ and 2.14 MPa, respectively. Zhang et al. obtained foam concrete with an average pore size smaller than 0.3 mm from silt and cement as a building backfill material [22]. Liu et al. investigated the effects of the fly ash content on the strength enhancement and pore structure of foam concrete, and the results showed that increasing the amount of fly ash could increase the average pore size [28]. Controlling the CO_2_ pressure and curing time can also improve the diffusion rate and sequestration efficiency of CO_2_. Many studies on the curing conditions have shown that the pressure and concentration of CO_2_ directly affect its diffusion capacity during the curing of foam concrete or ordinary concrete [29]. In general, the higher the CO_2_ pressure, the higher the sequestration of CO_2_. CO_2_ sequestration increases; however, the reaction rate decreases with curing time [30]. In addition, to sequester CO_2_ more efficiently, connected pores inside the material are highly desired. Studies have shown that it is completely feasible to prepare foam concrete with connected pores by optimizing the raw materials and reaction conditions. Zhang prepared a porous foam material with a tunable open pore structure (open pore 30–80%) using the ordinary PO and forming agent [31]. The material exhibited good pollutant adsorption performances.

Alkaline cementitious materials represented by cement have the ability to sequester CO_2_, yet most studies have focused on the effects of carbonization on the material strength. CO_2_ sequestration performance is less involved, and thus the pore-channel properties of the materials are rarely reported. In the present study, the effects of pore structure on CO_2_ sequestration performance were demonstrated. A series of foam concretes were prepared using PO cement at different water–cement ratios and different foaming agent dosages. This was performed by natural curing and CO_2_ curing using slag and fly ash as the raw materials and 30% hydrogen peroxide as the foaming agent. The materials were characterized by XRD, BET, SEM, and TGA to explore the influence mechanism of different systems on CO_2_ sequestration, aimed to provide guidance for the utilization of solid waste materials and mitigation of the greenhouse effect.

## 2. Experimental Design

Figure 1 shows the experimental flow chart. Firstly, different foam concrete materials were prepared by varying the water–cement ratio and the amount of foaming agent. The materials were cured for 24 h naturally or by CO_2_, demolded, dried to terminate hydration, ground, and characterized by TGA, XRD, BET, and SEM. CO_2_ sequestration was calculated from the TGA results to determine the optimal system. The hydration products, pore structure, specific surface area, and pore volume were analyzed by XRD, BET, and SEM to explore the influence mechanism of water–cement ratio and forming agent dosage on CO_2_ sequestration.

### 2.1. Preparation of Foam Concrete Materials

The cementitious material was prepared with 50% PO cement, 35% slag, and 15% fly ash. Hydrogen peroxide (30%) [32], sodium stearate, and calcium oxide were used as the foaming agent, foam stabilizer, and alkali activator, respectively. Table 1 lists the composition of each system.

The solid waste-based porous materials were obtained from the systems by chemical foaming. First, pour the pre-weighed cementing material and foaming agent dry powder material into the mixing pot for 1 min at low speed, and then pour the water at normal temperature into the mixing pot for 1 min at low speed according to the experimental requirements, add the CaO alkali activator and stir for 2 min. Finally, pour H_2_O_2_ into the mixing pot and mix with cement slag slurry, and quickly stir for 10s to prepare foamed concrete. After that, quickly pour it into the prepared 5 × 5 × 5 cm mold. After the foaming is finished, cover the surface with a layer of plastic wrap to prevent water evaporation. After standing for 1 day, remove the mold, wrap the sample in a plastic bag, and send it to the curing box for curing until the specified age (3 days). The actual picture of material preparation is shown in Figure 2.

### 2.2. CO_2_ Curing and Sequestration

The CO_2_ sequestration experiment simulated the real CO_2_ adsorption process in foam concrete. First, the curing chamber was purged with pure CO_2_ (99.9%) for a period of time to remove the air. The samples were placed in the curing chamber in the order of labels, and 0.1 MPa CO_2_ was continuously introduced for 24 h for CO_2_ curing and sequestration.

### 2.3. Characterizations

The hydration product composition of the cementitious material was determined by X-ray diffraction (XRD) using a Rigaku Ultima IV X-ray diffractometer in the scanning range of 5–80° at the scanning speed of 8°/min and the step size of 0.02°.

The evolution of the multiscale porous structure of the solid waste-based porous material with the variation of water–cement ratio and foaming agent dose was characterized by SEM and BET analysis. Specifically, the microstructure and surface morphology were imaged with a Nova Nano SEM 450 high-resolution scanning electron microscope, and the specific surface area and average pore diameter were measured with a fully automated ASAP-2460 surface area and porosity analyzer.

TGA was conducted on the prepared solid waste-based material samples at the heating rate of 10 °C/min under a nitrogen atmosphere. The pyrolysis process was analyzed with TGA and DTG curves, and CO_2_ sequestration was calculated from the weight loss and dry density of the sample.

## 3. Experimental Results and Analysis

### 3.1. Effects of Water–Cement Ratio on CO_2_ Sequestration

The naturally cured samples at the water–cement ratios of 0.4, 0.7, and 0.8 are denoted as NC 0.4, NC 0.7, and NC 0.8, respectively, and the corresponding carbon-cured samples are denoted as CC 0.4, CC 0.7, and CC 0.8, respectively. The hydration products, specific surface areas, average pore sizes, and CO_2_ sequestrations of these samples were analyzed and characterized to determine the optimal water–cement ratio.

#### 3.1.1. XRD Analysis

Figure 3 shows the XRD patterns of the naturally cured samples with different water–cement ratios. The diffraction peak at 2θ = 29.5° is the strongest, which is the characteristic peak of the main hydration product C-S-H, and the peak intensity increases with the increase of water–cement ratio. The diffraction peaks at 2θ = 36.9°, 47.1°, and 50.7° are attributed to Ca(OH)_2_. These diffraction peaks in NC 0.4 are not obvious, possibly because the low water–cement ratio causes incomplete hydration or low Ca(OH)_2_ content, which will eventually result in low CO_2_ sequestrations. The characteristic diffraction peaks of calcite are detected at 2θ = 29.4°, 39.4°, and 42.8°. The cementitious material contains calcite. CaCO_3_ can also be formed by the carbonization of Ca^2+^ during the material preparation. The diffraction peaks at 2θ = 25.7°, 26.6°, 32.1°, and 43.9° can be indexed to vaterite, and the peaks also become stronger with the increase of water–cement ratio. The diffraction peaks at 2θ = 10.9° and 23.3° can be assigned to the Ht phase, and that at 2θ = 30.8° is due to the mayenite phase. The comparison of the XRD patterns of samples with different water–cement ratios suggest they contain the same phases, indicating that changing the water–cement ratio will not alter the types of hydration products.

#### 3.1.2. SEM Analysis

Figure 4 shows the SEM images of the samples with different water–cement ratios. At the low water–cement ratio of 0.4, the samples exhibit dense surfaces, high dry densities, significantly ununiform pore sizes, wide size distributions, and small specific surface areas (Figure 4a1,a2). As the water–cement ratio increases, the sample surface becomes loose, the pores on the surface are refined, the dry density decreases, the number of pores in the sample increase, and pore sizes are more uniform (Figure 4b1,b2,c1,c2). Connected pores are clearly observed in NC 0.7 and CC 0.7 (Figure 4b1,b2), which are conducive to CO_2_ diffusion, increase the contact area between CO_2_ and foam concrete material and eventually improve CO_2_ sequestration. Their high magnification images reveal the fibrous hydration product C-S-H gel and the flaky Ca(OH)_2_ in NC0.7. However, there is much more CaCO_3_ in CC 0.7, and Ca(OH)_2_ flakes are barely seen (Figure 4d1,d2), suggesting most of the Ca(OH)_2_ is carbonated into CaCO_3_.

#### 3.1.3. BET Analysis

The samples NC 0.4, NC 0.7, NC 0.8, CC 0.4, CC 0.7, and CC 0.8 were then characterized for N_2_ adsorption/desorption isotherm and pore size distribution. As shown in Figure 5a, the nitrogen adsorption of all samples increases gradually up to a relative pressure of 0.5 and then increases dramatically as the relative pressure rises further. This pattern of nitrogen adsorption is due to the capillary condensation phenomenon, which is caused by the slit-like pore structure. The adsorption capacity is extremely high as the relative pressure is close to 1, approaching the saturation adsorption due to multi-layer adsorption. The N_2_ adsorption capacities before and after carbonization are different because the precipitates generated by the CO_2_ absorption block the mesopores in foam concrete and hinder the adsorption.

Figure 5b shows the pore size distributions of these samples, and Figure 6 compares their specific surface areas and pore structures. As can be seen, the pore sizes of all of them mainly fall between 2 and 15 nm, with some exceptions between 15 and 60 nm, indicating that the pores in the samples are mainly mesopores. Before carbonization, both pore volume and specific surface area gradually increase with the increase of water–cement ratio and reach the highest values of 0.144 cm^3^/g and 25.412 m^2^/g, respectively, at the ratio of 0.7. The average pore size shows similar changes, and the value at the water–cement ratio of 0.7 is 12.84 nm, second only to that of NC 0.8. Overall, CO_2_ diffusion is easier in NC 0.7. Its large surface area provides a large contact surface with CO_2_, which also improves CO_2_ sequestration.

#### 3.1.4. CO_2_ Sequestration

To determine the CO_2_ sequestration amount, TGA was conducted on the naturally cured and carbon-cured samples with different water–cement ratios. Figure 7 shows their TGA and DTG curves. All samples show constant weight losses during the whole heating process. The weight loss between 30 °C and 100 °C is mainly caused by the evaporation of adsorbed water, and that from 100–200 °C is mainly due to the decomposition of the hydration product C-S-H. The decomposition of the hydration product Ht phase results in the weight loss between 200 °C and 350 °C. The weight loss between 350 °C and 500 °C is mainly caused by the loss of Ca(OH)_2_, and that between 500 °C and 900 °C is mainly attributed to the decomposition of carbonate. The carbonate is mainly from the material itself and the carbonization of Ca(OH)_2_. Therefore, CO_2_ sequestration can be obtained as the difference in the weight loss between naturally cured and carbon-cured samples during the carbonate decomposition.

As can be seen from Figure 7a, in the naturally solidified sample, the weight loss rate of NC 0.8, NC 0.7, and NC 0.4 is 19.59%, 17.55%, and 18.75%, respectively, within the whole temperature range. As shown in Figure 7b, the weight loss rate of NC 0.4 is 4.62%, that of NC 0.7 is 4.03%, and that of NC 0.8 is 4.15% in the temperature range of 500–900 °C of carbonate decomposition, which is mainly the carbonation of Ca(OH)_2_ in the material itself.

Specifically, it shows the highest weight losses from 30–100 °C and 200–350 °C, corresponding to the decomposition of the Ht phase (Figure 7b). The weight loss of NC 0.4 is the highest in the Ca(OH)_2_ decomposition temperature range, indicating that its Ca(OH)_2_ consumption is the least. The weight loss of NC 0.4 in the carbonate decomposition temperature range of 500–900 °C is higher than those of the other two naturally cured samples.

Among the carbon-cured samples, the weight loss of CC 0.7 is the highest, with values up to 28.25% (Figure 7c). The weight loss rate of CC 0.4 is 27.05%, and that of CC 0.8 is 26.62%. As shown in Figure 6d, the weight loss rate of CC 0.4 is 18.07%, and that of CC 0.7 is 19.39% in the 500–900 °C temperature range of carbonate decomposition. The weight loss rate of CC 0.8 is 19.87%, which is mainly due to the effective absorption of CO_2_ by all hydration products in the carbonization process and their conversion into CaCO_3_-dominated carbonates.

Unlike the naturally cured sample, CC 0.7 shows the highest weight loss in the temperature range of 500–900 °C among its different weight loss temperature ranges. As can be seen from the DTG curves of the carbon-cured samples, they exhibit the same weight loss temperature ranges as the naturally cured samples; however, the weight losses in the decomposition temperature ranges of Ca(OH)_2_ and Ht phase are lower, and the weight loss in the carbonate decomposition temperature range of 500–900 °C is much greater (Figure 7d). These results indicate that during CO_2_ curing, the hydration products and CO_2_ are converted into carbonates. The weight loss of carbonate increases first and then decreases with the increase of water–cement ratio, and so does the sequestration content of CO_2_, indicating that extremely high water–cement ratios are inconducive to CO_2_ sequestration. The high dry density, dense structure, fewer pores, and ununiform pore sizes of the material with the water–cement ratio of 0.4 are unfavorable to CO_2_ sequestration. At the water–cement ratio of 0.7, the strong hydration reaction of the cementitious material results in a loose structure, many uniform pores, and even connected pores, which can promote CO_2_ diffusion between the pores and increase the contact area. Further increasing the water–cement ratio to 0.8 decreases CO_2_ sequestration, which may be due to the good fluidity of the slurry, which easily breaks the bubbles and reduces the pore volume. Although more hydration products are generated, the reductions in CO_2_ diffusion rate and contact area cause low sequestration.

### 3.2. Effects of Foaming Agent Dosage on CO_2_ Sequestration

The study on the effects of water–cement ratio on CO_2_ sequestration suggests that the optimal water–cement ratio is 0.7. Therefore, the effects of the foaming agent dosage on CO_2_ sequestration were investigated with the water–cement ratio fixed at 0.7 and foaming agent dose varied to 8%, 10%, and 12%, respectively. The corresponding naturally cured samples are denoted as NC 8%, NC 10%, and NC 12%, and the corresponding carbon-cured samples are denoted as CC 8%, CC 10%, and CC 12%. The hydration products, specific surface areas, average pore sizes, and CO_2_ sequestrations of these samples were analyzed and characterized to determine the optimal foaming agent dosage.

#### 3.2.1. XRD Analysis

Figure 8 shows the XRD patterns of the samples prepared at different foaming agent doses. As can be seen, the changes in the XRD pattern with the foaming agent dosage are similar to those with the change in water–cement ratio. The diffraction peak at 2θ = 29.5° is the strongest one, which is the characteristic diffraction peak of the main hydration product C-S-H of the cementitious material. The intensity of the diffraction peak increases with the increase of the foaming agent dosage. The diffraction peaks at 2θ = 36.1°, 34.1°, 47.1°, and 52.2° are indexed to Ca(OH)_2_. These diffraction peaks also become stronger with the increase of foaming agent dosage. The diffraction peaks at 2θ = 10.9° and 23.3° are indexed to the Ht phase, which is a double-layer of metal hydroxides composed of Mg^2+^ (or Ca^2+^) and Al^3+^. Similarly, the diffraction peaks of CaCO_3_ at 2θ = 29.4°, 39.4°, and 43.2° are observed in the XRD patterns of all samples, suggesting that main phases remain unchanged with the increase of foaming agent dosage and the foaming agent dosage has no effects on the type of hydration products of the material.

#### 3.2.2. SEM Analysis

The SEM imaging reveals that the foaming agent dosage affects the microscopic pore distribution in foam concrete significantly (Figure 9). At the foaming agent dose of 8%, more round pores are formed in the sample; however, pore sizes are ununiform. The low foaming agent dose also results in a low relative water content and dense structure. As the foaming agent dose increases to 10%, the relative water content and the degree of hydration of the cementitious material increase, and more bubbles are generated. There are more round pores between the particles of the cementitious material, and connected pores are clearly observed, which lowers the resistance to CO_2_ diffusion, increases the contact area, and thus is conducive to CO_2_ sequestration. However, further increasing the foaming agent dose to an excess amount, 12%, the number of bubbles increases, and the bubbles tend to break or connect with each other to form bigger bubbles, which leads to low dry density and thus is inconducive to CO_2_ sequestration. It is worth noting that the hydration products C-S-H gel and Ca(OH)_2_ flakes are also observed in the 5000× magnification image.

#### 3.2.3. BET Analysis

The pore structure of foam concrete is an important factor affecting CO_2_ sequestration efficiency. In addition to water–cement ratio, foaming agent dosage is also an important factor affecting the pore structure. Therefore, NC 8%, NC 10%, NC 12%, CC 8%, CC 10%, and CC 12% were characterized by BET analysis for their pore properties. Figure 10 and Figure 11 show the N_2_ adsorption/desorption isotherms, pore size distributions, and specific surface areas of these samples.

Similar to those obtained at different water–cement ratios, the N_2_ adsorption/desorption isotherms of the samples prepared with different foaming agent dosages are the typical type IV isotherms (Figure 10a). The N_2_ adsorption capacity increases with the increase of pressure, indicating that the pore structure is still dominated by mesopores, despite the variations in the foaming agent dosage.

The pore sizes range from 2 nm to 70 nm and mainly distribute in the range of 3–10 nm (Figure 10b). Although the specific surface area of NC10% is slightly smaller, its pore volume and average pore size are the largest, which is conducive to CO_2_ diffusion and transport, improves the reaction efficiency between CO_2_ and the hydration products or Ca^2+^, and thus increases CO_2_ sequestration.

#### 3.2.4. CO_2_ Sequestration

CO_2_ sequestration directly reflects the CO_2_ adsorption performance of a foam concrete material. Therefore, TGA analysis was also carried out on the foam concrete materials prepared with different foaming agent doses to calculate their CO_2_ sequestrations. As shown in Figure 12a,c, each sample shows obvious weight loss in the temperature range of 30–900 °C. The weight loss of NC12% is the highest among the naturally cured samples, with values of up to 23.8%. The weight loss rate of 10% NC is 17.55%, and that of 8% NC is 23.64. CC 10% has a maximum weight loss of 28.21%, CC 12% has a weight loss of 24.37, CC 8% has a weight loss of 23.98%, and that of CC 10% is the highest among the carbon-cured sample with values of up to 27.0%. DTG curves suggest that the weight loss caused by the decomposition of the Ht phase at 200–350 °C and that of Ca(OH)_2_ at 350–500 °C are lower in the carbonized sample than in the naturally cured sample (Figure 12d vs. Figure 12b). However, in the temperature range of 500–800 °C, the weight loss of the carbon-cured sample is higher, suggesting that the hydration products effectively absorb CO_2_ and convert it into CaCO_3_. The weight loss caused by the decomposition of carbonate gradually increases with the increase of the foaming agent dose and then declines at the dose of 12%. This can be explained by the fact that the relative water content of the material increases with the increase of the foaming agent dose, and the hydration reaction of the material becomes more thorough. At the same time, the number of generated bubbles increases, which provides more adsorption sites for CO_2_ sequestration. However, the extremely high amount of foaming agent tends to generate large pores. The good fluidity of the slurry also tends to break the foam in the concrete and thus reduces the pore volume, which leads to the reduction of the actual adsorption site for CO_2_ and lowers CO_2_ sequestration.

## 4. Discussion

There have been many studies on the recovery, capture, and utilization of CO_2_; however, most methods have problems of poor storage effect and high cost. In this study, industrial wastes such as cement, slag, and fly ash (in combination with hydrogen peroxide) were used as chemical foaming agents in the production of foamed concrete. This was performed using an alkali excitation agent, and the pore structure of the material was adjusted by changing the water–cement ratio and the amount of foaming agent. It was found that C-S-H, Ca(OH)_2_, and other substances produced in the hydration process of the material would precipitate carbonate with carbon dioxide. With the increase of water–cement ratio and dosage of foaming agent, thermogravimetric data showed that the carbon dioxide sealing capacity showed a trend of first increasing and then decreasing. It was found by BET and SEM that when the water–cement ratio increased to 0.7, and the dosage of foaming agent increased to 10%, the specific surface area of the material gradually increased, the proportion of connected pores increased, and the size of pores gradually became uniform. The samples are all mesoporous, with pore sizes ranging from 2–50 nm. With the further increase of water–cement ratio and dosage of foaming agent, the sealing effect is poor because the foaming is too intense, the water–cement ratio is high, and bubbles burst easily, leading to the increase of the proportion of large pores. After CO_2_ storage, the specific surface area and average pore diameter of the sample increased slightly, which could be caused by the hydration reaction of Ca^2+^ on the surface of slag particles before carbonization. During the carbonization process, Ca^2+^ was generated by continuous decalcification of C-S-H gel, and CO_2_ was dissolved into CO_3_^2−^ and HCO_3_^−^ in the carbonization chamber. Ca^2+^ combines with CO_3_^2−^ in the solution to produce CaCO_3_ with a stable structure. After decalcification, C-S-H gels undergo condensation reactions with each other and finally form calcium-modified gels with a higher degree of polymerization, forming deeper ion-leaching channels, thus increasing pore volume and average pore diameter.

This study explored the optimal water–cement ratio and amount of foaming agent. The amount of alkali activator and alkali excitation system are also important factors affecting the adsorption of carbon dioxide. In the next step, we will continue to research the influence of the amount and type of alkali activator on the pore structure of foamed concrete materials and the amount of carbon dioxide adsorption.

## 5. Conclusions

In this paper, foam concrete materials were prepared from cement and solid waste-based materials using hydrogen peroxide as the foaming agent. The effects of water–cement ratio and foaming agent dosage on CO_2_ adsorption were studied by XRD, BET, SEM, and TGA. The following conclusions have been drawn.

(1) XRD analysis reveals that the hydration products of the cementitious material mainly include C-S-H, Ht, and Ca(OH)_2_, and they are responsible for CO_2_ sequestration. The types of hydration products remain the same, despite the changes in water–cement ratio and foaming agent dosage.

(2) BET analysis suggests that the pores in the foam concrete materials are mainly mesopores. The specific surface area, pore volume, and average pore size of the material change with the changes in water–cement ratio and foaming agent dosage. When the water–cement ratio, foaming, and dosage increase continuously, the specific surface area of foamed concrete materials increases first and then decreases. When the water–cement ratio is 0.7, the specific surface area reaches the maximum, increasing the contact area between CO_2_ and materials and increasing the adsorption point, which is conducive to the solidification of CO_2_.

At the water–cement ratio of 0.7 and the foaming agent dose of 10%, the obtained foam concrete exhibits a specific surface area of 25.412 m^2^/g, a pore volume of 0.144 cm^3^/g, and an average pore diameter of 12.840 nm.

(3) SEM analysis shows that the hydration degree of the cementitious material is low at low water–cement ratios and low foaming agent dosages. The surface structure of the obtained sample is dense, with fewer pores and ununiform big pores. At the water–cement ratio of 0.7 and foaming agent dose of 10%, more pores with uniform sizes are formed, and even connected pores are observed, which is conducive to the CO_2_ diffusion in the material.

(4) CO_2_ sequestration can be determined from the TGA and DTG curves. The decompositions of the hydration products or the carbonate produced from carbonization of Ca(OH)_2_ occur in the temperature range of 550–850 °C. CO_2_ sequestration first increases and then decreases with the increases of water–cement ratio and foaming agent dosage. The highest CO_2_ sequestration of 66.35 kg/m^3^ is obtained at the water–cement ratio of 0.7 and the foaming agent dose of 10%.

Through the CO_2_ curing experiment of foamed concrete materials prepared under the conditions of water–cement ratio and foaming agent, it is concluded that when the water–cement ratio is 0.7 and foaming agent dosage is 10%, the foamed concrete materials have uniform pore size, high porosity and connected pore ratio, large specific surface area of pores, and the highest CO_2_ sealing stock is 66.35 kg/m^3^. Therefore, the water–cement ratio is 0.7. The foaming agent content of 10% is the best proportion in this group of raw material systems.

## Figures and Tables

**Figure 1 materials-16-03441-f001:**
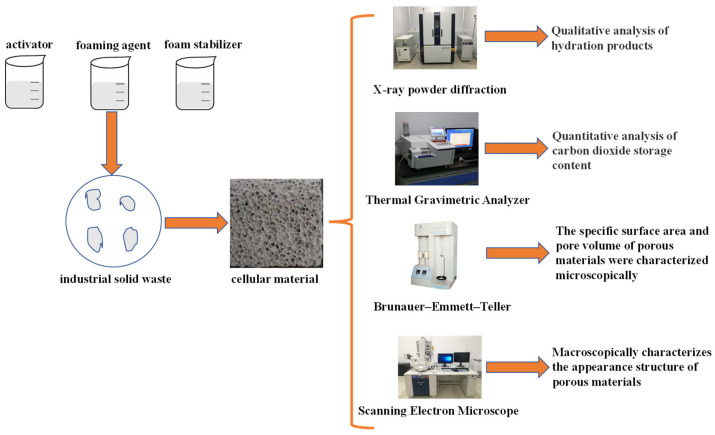
Experimental flow chart.

**Figure 2 materials-16-03441-f002:**
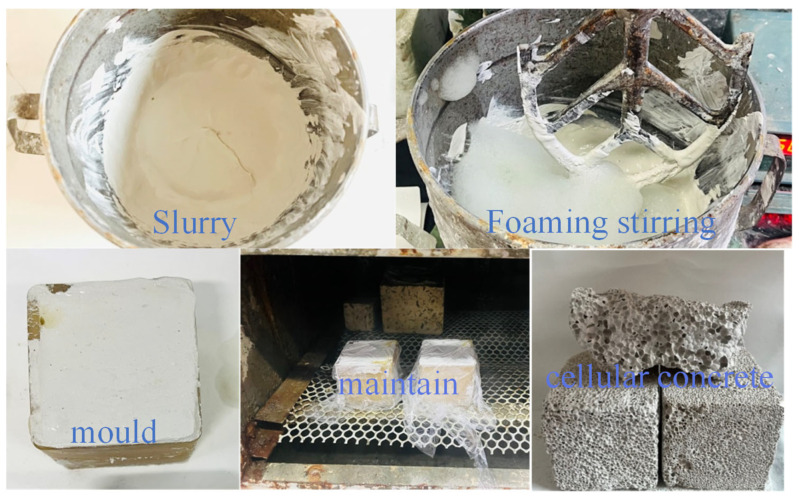
Actual picture of material preparation.

**Figure 3 materials-16-03441-f003:**
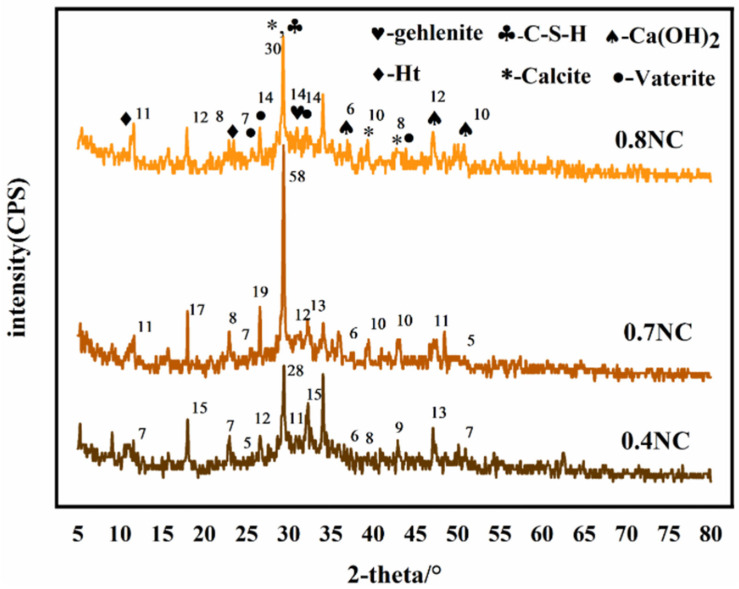
XRD patterns of the samples with different water–cement ratios.

**Figure 4 materials-16-03441-f004:**
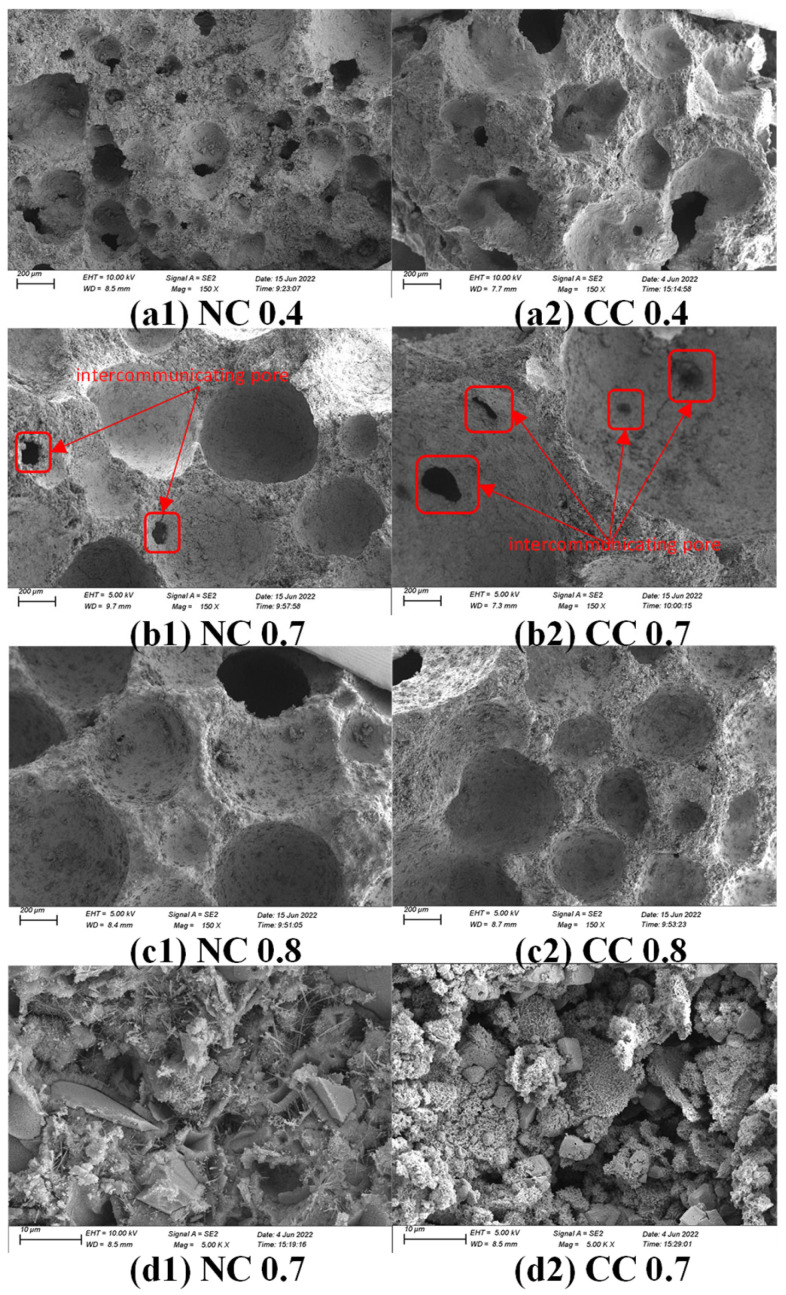
SEM images of samples with different water–cement ratios before and after carbonization and high magnification images of the samples with water–cement ratio of 0.7 (**a1**) water-cement ratio 0.4, non-carbonized sample with 10% foaming agent; (**a2**) water-cement ratio 0.4, carbonized sample with 10% foaming agent; (**b1**) water-cement ratio 0.7, non-carbonized sample with 10% foaming agent; (**b2**) water-cement ratio 0.7, carbonized sample with 10% foaming agent; (**c1**) water-cement ratio 0.8; For non-carbonized sample with 10% foaming agent, (**c2**) water-cement ratio 0.8, and carbonized sample with 10% foaming agent, (**d1**) water-cement ratio 0.7 at 5000 times magnification, and (**d2**) sample with 10% foaming agent after carbonization at 5000 times magnification.

**Figure 5 materials-16-03441-f005:**
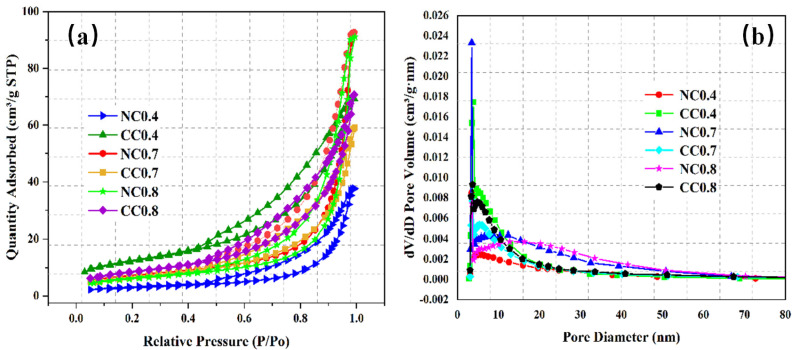
N_2_ adsorption/desorption isotherms (**a**) and pore size distributions (**b**) of samples with different water–cement ratios.

**Figure 6 materials-16-03441-f006:**
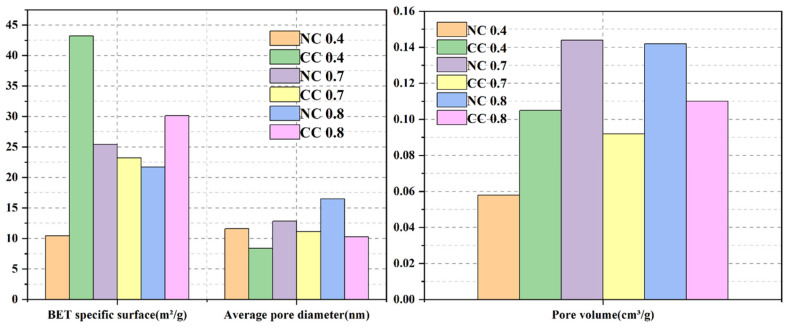
Comparison of the specific surface areas and pore structure properties of the samples with different water–cement ratios.

**Figure 7 materials-16-03441-f007:**
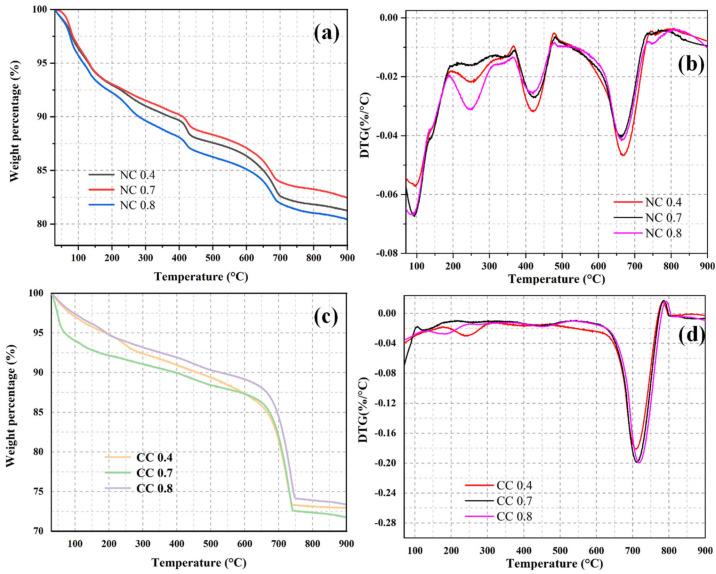
GA curves (**a**,**c**) and DTG curves (**b**,**d**) of samples with different water–cement ratios.

**Figure 8 materials-16-03441-f008:**
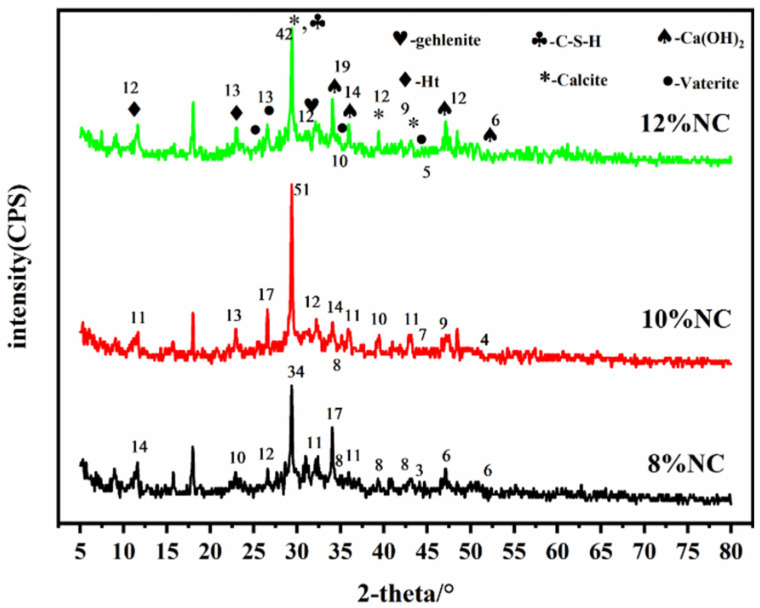
XRD patterns of the samples prepared with different foaming agent doses.

**Figure 9 materials-16-03441-f009:**
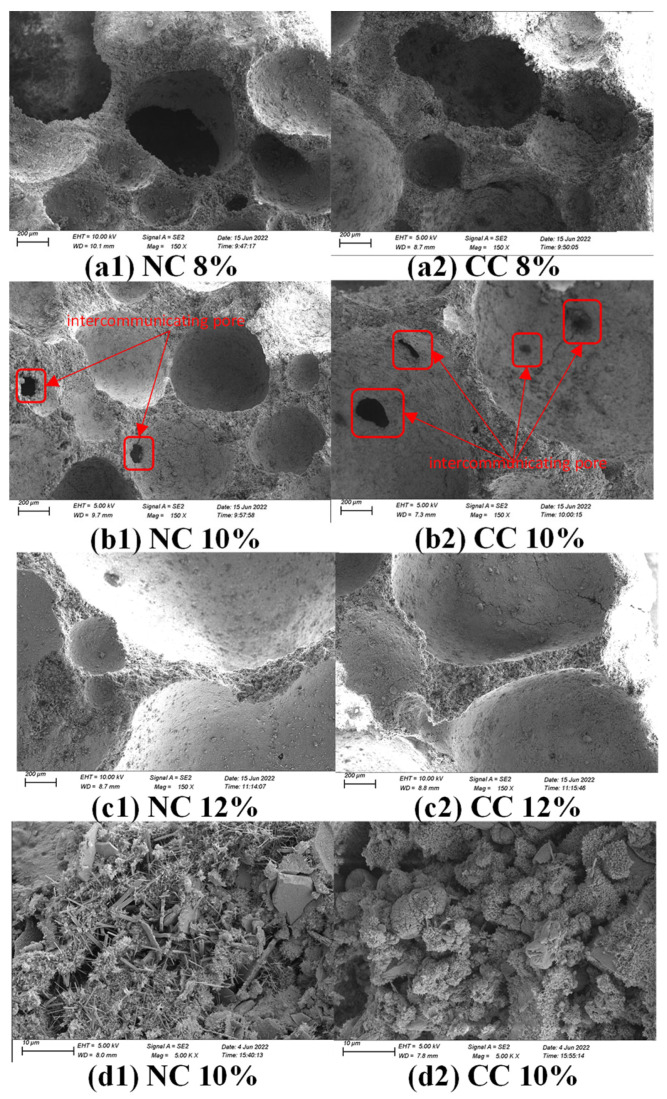
SEM images of samples prepared with different foaming agent doses before and after carbonization and high magnification images of the samples prepared with 10% foaming agent. (**a1**) uncarbonized sample with a water-cement ratio of 0.7 and foaming agent of 8%, (**a2**) uncarbonized sample with a water-cement ratio of 0.7 and foaming agent of 8%, (**b1**) uncarbonized sample with a water-cement ratio of 0.7 and foaming agent of 10%, (**b2**) water-cement ratio of 0.7 and foaming agent of 10% and (**c1**) water-cement ratio of 0.7, For the uncarbonized sample with 12% foaming agent, (**c2**) water-cement ratio 0.7, and the carbonized sample with 12% foaming agent, (**d1**) water-cement ratio 0.7, and the uncarbonized sample with 10% foaming agent, and (**d2**) 5000-fold magnification of water-cement ratio 0.7, and the carbonized sample with 10% foaming agent.

**Figure 10 materials-16-03441-f010:**
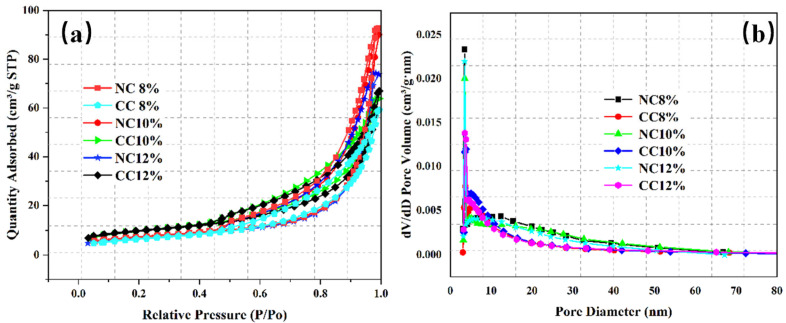
N_2_ adsorption/desorption isotherms of the samples prepared with different foaming agent dosages before and after carbonation (**a**) and their pore diameter distributions (**b**).

**Figure 11 materials-16-03441-f011:**
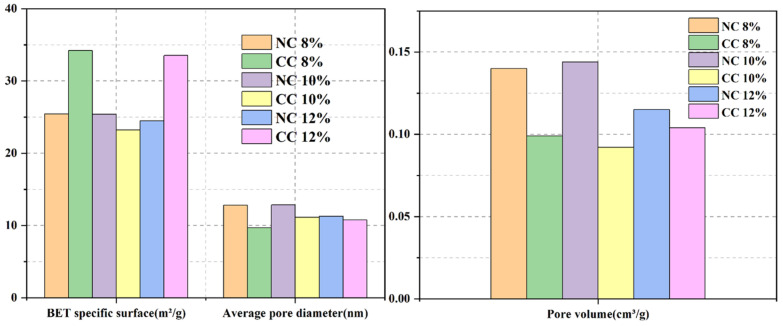
Comparison of specific surface areas and pore structures of the samples prepared with different foaming agent dosages.

**Figure 12 materials-16-03441-f012:**
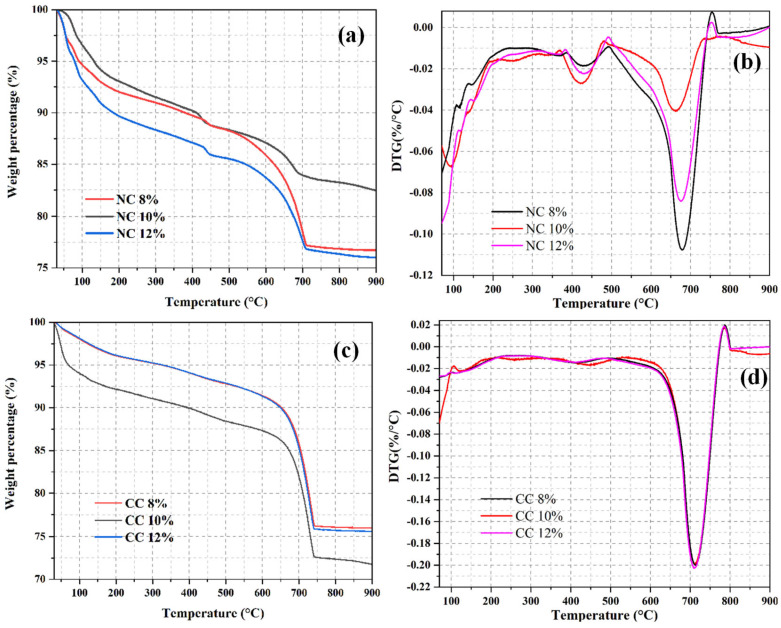
TGA (**a**,**c**) and DTG (**b**,**d**) curves of the samples prepared with different foaming agent doses.

**Table 1 materials-16-03441-t001:** Foam concrete mix ratios.

Cement	Slag	Fly Ash	Water–Cement Ratio	30% Hydrogen Peroxide	Calcium Oxide	Sodium Stearate
50%	35%	15%	0.4	10%	5%	0.8%
			0.7			
			0.8			
			0.7	8%		
				12%		

## Data Availability

The data used for conducting classifications are available from the corresponding author upon request.

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
