# Peer review of "Study on Carbon Fixation Ratio and Properties of Foamed Concrete"

_materials, 2023, doi:10.3390/ma16093441_

Round 1

Reviewer 1 Report

1. Table 1 put in order to improve the perception of each composition

2. Figure 6 is missing in the text of the article, there is only a description of it.

3. In Figure 11, it is necessary to graphically note the change in the mass of samples during heating in%, references in the text to the concepts of more or less are incorrect for a scientific article. Figure 6 should be done in the same way.

4. In Figure 3, the images of the microstructure (b NC 0,7 and B? CC 0.7) are duplicated by the images in Figure 8 (b NC 10% and B? CC 10%). In addition, why in one case there are different letters in the designations - uppercase and uppercase.

5. The spectrum of the XRD in Figure 2 (0.7 NS) copies the spectrum XRD in Figure 7 (8% NC)

Reviewer 2 Report

The manuscript entitled “Study on Carbon Fixation Ratio and Properties of Foamed Concrete” has addressed an interesting topic. I was well written in terms of language. The following comments may help the author to enhance the manuscript: 

1)    The result and finding should be highlighted with a brief discussion. 

2)    In section (2.) The details of preparation of famed agents must be stated for Example the ratio of water to foamed agent, the type of foamed agent used, also support the methodology with references. 

3)    Add subtitles to explain in detail mixture proportion and specimen preparation.

4)    The results need to be discussed with supporting evidence from previous studies.

5)    The values of intensity in XRD results need to be added in Y-axis.

6) Some references are below 2010. It will be better if you update the references. 

Reviewer 3 Report

1- abstract need to re-write. Add novelty it in. Add increase or decrease behavior of results. 

2- In introduction section, add latest literature review. Add new papers and in last paragraph, add your original contribution.

3- in experimental design section, add actual pictures of materials and specimen casting and testing procedures.

4- in fig. 2 legends are not clear 

5- in fig. 5 and 10, what are the units of y axis. Moreover, difference in those results are significant?

6- write specific conclusions.

Round 2

Reviewer 1 Report

1. Table 1 has not been edited in accordance with my comments

2. The microstructure in figures 4 and 9 is duplicated (b and B NC10% and b and B NC 0.7% ). Why are the symbols on the left in lowercase letters, and on the right in capital letters?

Reviewer 3 Report

Previous comments were addressed

Author Response

Thank you for your criticism, correction and approval of our paper.